# Recent Advances in Polydopamine for Surface Modification and Enhancement of Energetic Materials: A Mini-Review

**Ziquan Qin** [1], **Dapeng Li** [2], **Yapeng Ou** [1,*] , **Sijia Du** [1], **Qingjie Jiao** [1], **Jiwu Peng** [3] **and Ping Liu** [3]

[1] State Key Laboratory of Explosion Science and Technology, Beijing Institute of Technology, Beijing 100081, China
[2] Xi'an Modern Chemistry Research Institute, Xi'an 710065, China
[3] Chongqing Hongyu Precision Industry Group Co., Ltd., Chongqing 402760, China
* Correspondence: ouyapeng@bit.edu.cn

**Abstract:** Polydopamine (PDA), inspired by the adhesive mussel foot proteins, is widely applied in chemical, biological, medical, and material science due to its unique surface coating capability and abundant active sites. Energetic materials (EMs) play an essential role in both military and civilian fields as a chemical energy source. Recently, PDA was introduced into EMs for the modification of crystal phase stability and the interfacial bonding effect, and, as a result, to enhance the mechanical, thermal, and safety performances. This mini-review summarizes the representative works in PDA modified EMs from three perspectives. Before that, the self-polymerization mechanisms of dopamine and the methods accelerating this process are briefly presented for consideration of researchers in this field. The future directions and remaining issues of PDA in this field are also discussed at last in this mini-review.

**Keywords:** polydopamine; self-polymerization; energetic materials; interface enhancement; crystal transformation inhibition



## 1. Introduction

In 2007, Lee et al. proposed the surface chemistry for multifunctional coatings, which is inspired by the main component of adherent proteins in mussel, namely, dopamine [1,2]. Dopamine is able to form adhesive polydopamine (PDA) thin films on various material surfaces by oxidative self-polymerization in an alkaline aqueous medium. Fascinatingly, PDA films can also provide an important platform for secondary reactions since there are many reactive groups, such as catechol, amine, and imine, on its surface, which can serve as the starting points for covalent modification with desired molecules [3].

Due to its unique surface-coating capability and abundant active sites, unsurprisingly, PDA was rapidly applied in many fields across the chemical, biological, medical, and material sciences [4]. In barely more than a decade, the physicochemical properties of PDA were extensively studied, covering its biocompatibility and biodegradation, electrical conductivity, metal ions chelating and redox activities, and other potential properties. Among all these attractive features, its adhesive property and chemical reactivity are closely related, while they are also the basis of the vast majority of applications [5,6]. The adhesive properties of different mussel foot proteins with diverse compositions and contents of 3,4-dihydroxyphenyl-L-alanine (DOPA, a derivative of PDA) were investigated, and it was concluded that possible interaction mechanisms between DOPA and substrates involve electrostatic, hydrogen bonding, hydrophobic interactions, cation–π, π–π stacking, and metal complexation [7]. Further, the roughness of the PDA nanomembrane was also observed as a factor deciding its adhesion [8]. The abovementioned generalizable theories and trends were critical for PDA coating and surface functionalization in engineering practices.

Energetic materials (EMs) are very important power resources for civilian and military applications, covering explosives, propellants, and pyrotechnics, which release energy

in the forms of combustion and explosion. PDA was introduced into EMs until very recently. Lin et al. fabricated a very compact PDA film on the surface of an explosive crystal, octogen (HMX), which not only improved the stability of the crystal phase, but also synergistically enhanced the mechanical, thermal, and safety performances of HMX-based polymer-bonded explosives (PBXs) [9]. Meanwhile, Yan's group was interested in the control of the reactivity of metastable intermixed composites (MICs) by constructing a PDA interfacial layer between the nanoscale fuel and oxidizer [10], which resulted in an increased energy release and reduced sensitivity.

This review will summarize the recent progress of PDA surface modification for EMs. In the first section, the self-polymerization mechanisms and some methods to accelerate this polymerization process will be briefly discussed. The applications of PDA in EMs are presented in the main section, in terms of its functions including crystal transformation inhibition, interfacial bonding in composites, and related mechanical enhancement, as well as thermal stability and sensitivity modification. A straightforward summary of this review and some existing problems will be given in the last section. We hope this review will promote the application of PDA coating in EMs after solving some concerning issues.

## 2. Self-Polymerization Mechanisms and Catalytic Reactions of PDA

The self-polymerization of dopamine and deposition of PDA is a very facile but time-consuming process. A typical procedure may need 12 h including the dissolution and polymerization of the dopamine monomer (commercially, dopamine hydrochloride is typically used) in an alkaline aqueous solution (Tris-HCl buffer with pH 8.5 is typically used) without any sophisticated operations, presenting a significant color change from colorless to deep brown.

Although PDA can be facilely and mildly fabricated, its reaction mechanisms remain controversial due to the complex redox process and related intermediates during polymerization. The polymerization process was speculated as the oxidation of dopamine to dopamine-quinone, followed by intramolecular cyclization, oxidization, and rearrangement to form 5,6-dihydroxyindole (DHI) [11]. Furthermore, DHI and its oxide can eventually form a cross-linked polymer through the reverse dismutation reaction between catechol and o-quinone. Although the cyclized, nitrogenous species such as the indole- or indoline-type structures were also confirmed by other researchers, a distinct model was proposed that PDA was considered to be an aggregate of monomers cross-linked primarily via strong, noncovalent forces [12]. Hong et al. suggested that the formation of PDA is the combination of noncovalent self-assembly and covalent polymerization [13]. They identified a dual-path formation process, in which both paths form the oxidative product of dopamine, DHI, since a physical, self-assembled trimer of (dopamine)$_2$/DHI was observed by HPLC. Instead, Alfieri et al. suggested that PDA's polymerization mechanisms might have an alternative pathway besides the conventional DHI-oligomers as the essential intermediate [14]. Their experiments found that dissolved DHI cannot form PDA; rather, dopamine polymerization, on mechanism-based analysis, may arise by quinone-amine conjugation leading to polycyclic systems with extensive chain breakdown. It seems that the molecular mechanisms behind the polymerization and deposition of PDA are quite complicated and still unclear.

One part is for certain, though; oxidation plays a decisive role in PDA formation, which provides meaningful inspiration for accelerating this process. Several methods were excogitated to overcome the drawback of the low polymerization rate, for instance, UV irradiation, electrochemical actuation, and oxidant promotion [15–19]. Among all this research, Zhang et al. used $CuSO_4/H_2O_2$ to induce the polymerization of dopamine and accelerate the deposition rate of PDA [20,21]. Their works obtained a uniform PDA thin film with a thickness of 30 nm in 0.7 h, which is a considerably rapid deposition rate compared to other works, as shown in Table 1. Another attempt to achieve rapid PDA formation included the rising reaction temperature from the perspective of kinetics, which fabricated a PDA film in 0.5 h and obtained similar properties as those polymerized by the conventional method in 24 h [22], but the surface of the PDA obtained by this method

was relatively rough due to the shambolic deposition of PDA nanoparticles under vigorous stirring. Interestingly, the self-polymerization of dopamine normally occurs in alkaline conditions; however, some researchers also explored the possibility of PDA formation in an acidic environment [23,24], on the basis of understanding the relevance of oxidation.

**Table 1.** Thickness and deposition rate of PDA coatings with different methods.

| Methods/Conditions | Time (h) | Thickness (nm) | Deposition Rate (nm/h) | Ref. |
|---|---|---|---|---|
| Air, pH 8.5 | 24.0 | 50.0 | 2.1 | [2] |
| Pure $O_2$, pH 8.5 | 0.5 | 4.4 | 8.8 | [17] |
| UV, pH 8.5 | 2.0 | 4.0 | 2.0 | [15] |
| 0.5 V, pH 6.0 | 1.0 | 11.5 | 11.5 | [16] |
| AP, pH 7.0 | 2.0 | 70.0 | 35.0 | [18] |
| $NaIO_4$, pH 5.0 | 2.0 | 45.0 | 22.5 | [19] |
| $CuSO_4/H_2O_2$, pH 8.5 | 0.7 | 30.1 | 43.0 | [20] |

Understanding the potential mechanisms of accelerating the polymerization and deposition rates of PDA is purposeful to engineering practices, not limited to EMs. However, given that most of EMs having oxidability, PDA formation on the surface of EMs seems to be rational and logical. For example, Ammonium perchlorate (AP), which is a commonly used oxidant in composite solid propellants, was employed to induce polymerization of dopamine in Wei et al.'s work [18]. Further, Lin et al. also realized the limitation of the low polymerization rate and inaccurate kinetic control of PDA in EMs, and they investigated the kinetics of PDA formation under different conditions, including concentrations of dopamine and oxygen, as well as environment temperatures, when studying the PDA coating on an insensitive high explosive 1,3,5-triamino-2,4,6-trinitrobenzene (TATB) [25].

## 3. Applications of PDA in Energetic Materials

The applications of PDA in EMs mainly include the compact coating and surface functionalization, which are able to improve the stability of crystals, the reactivity of composites, as well as the mechanical, thermal, and safety properties, respectively.

### 3.1. Surface Modification of Energetic Crystals

Gong's group firstly introduce PDA coating into the field of EMs when they modified the crystal phase stability of a polymorphic high explosive, 1,3,5,7-tetranitro-1,3,5,7-tetraazacyclooctane (HMX), as shown in Figure 1. In the referenced work, the crystal transformation temperature of HMX (from $\beta$ to $\delta$ phase) was improved by 28 °C with PDA of 0.5 wt.%, and mechanical sensitivity was also reduced after thermal damage compared with raw HMX [26]. Their following work also investigated the kinetics of the polymorphic transition of PDA-coated HMX and probed the possible inhibition mechanisms basing on the density functional theory calculation [27], which suggested that PDA coating significantly decreased the polymorphic transition rate, especially for the nucleation process.

2,4,6,8,10,12-Hexanitro-2,4,6,8,10,12-Hexaazaisowurtzitane (HNIW, CL-20) is the most powerful military explosive and has promising application prospects. However, there are four crystal phases for CL-20 at ambient temperature, including $\alpha$, $\beta$, $\gamma$ and $\varepsilon$ phases, which can transfer into other phases under certain conditions. Accordingly, Jiao's group fabricated a stable core–shell CL-20/PDA structure by the conventional PDA formation method, which not only increases the $\varepsilon$-CL-20 crystal transformation temperature by about 30 °C, but also significantly improves the mechanical sensitivity and thermal stability [28]. PDA-coated graphene oxide was used to dope $\varepsilon$-CL-20 in situ during its crystallization in Huang's work [29]. The obtained CL-20 has a polygon shape with a smooth surface and relatively small size, as well as an improved polymorphic transition temperature up to 19 °C, and with some special dopants, a completely new crystal phase was even observed except in the four reported phases.

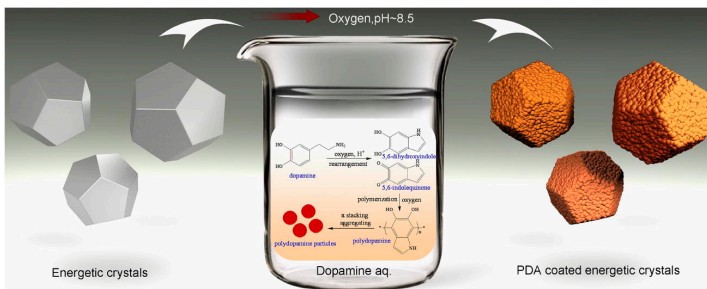

**Figure 1.** The successful introduce of dopamine chemistry in an energetic system as reported firstly: compact core–shell structure for every single energetic crystal with highly enhanced thermal stability. Reprinted with permission from Ref. [26]. 2017, Elsevier.

Lin extended the application of PDA from HMX to TATB for modifying its irreversible thermal expansion and mechanical properties by constructing a microcapsule [30,31]; due to the highly cross-linked and dense PDA shell with 1.5 wt.%, the irreversible expansion strain at room temperature dropped from 0.520% to 0.376%, and the tensile strength and toughness of TATB/PDA composites were 73% and 219% higher, respectively, than that of pristine TATB. Thenceforward, the mechanical enhancement of EMs by PDA and mechanisms behind this effect attracted more research interest.

### 3.2. Interfacial Bonding and Mechanical Enhancement of Energetic Composites

He's group fabricated a dense PDA film with a thickness of 56 nm by adjusting the reaction time, pH value, and temperature, and used PDA-coated TATB as the solid filler of a polymer-bonded explosive (PBX) [32]. The abovementioned PBX exhibited significantly improved tensile and compression strength or strain and creep resistance, due to the strong interfacial interaction between crystalline particles and binders with PDA as the interlayer, as shown in Figure 2. Moreover, they proposed a new model combining the chemical "interactional bonding" and physical "interlocking block" at the interface to explain the enhancement of mechanical properties which are demonstrated as hydrogen bonds and π–π interactions between the PDA and polymer binder, and increased roughness on the crystal surface, respectively.

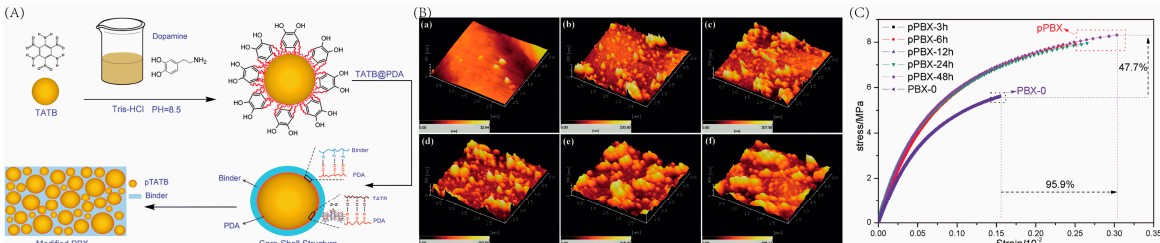

**Figure 2.** (**A**) Preparation of core-shell structured pTATB and the supposed interaction between PDA, TATB, and the fluoropolymer. (**B**) Topographical AFM images of PDA films deposited on the TATB crystal for (**a**) 0 h, (**b**) 3 h, (**c**) 6 h, (**d**) 12 h, (**e**) 24 h, (**f**) 48 h. (**C**) Stress-strain curves of PBX composites prepared from the abovementioned TATB. Reprinted with permission from Ref. [32]. 2012, Royal Society of Chemistry.

According to He's model, Lin designed and fabricated PDA-coated HMX, TATB, and 2,6-diamino-3,5-dinitropyrazine-1-oxide (LLM-105), and used these modified explosive crystals in PBXs [33,34]. The compressive and tensile strength of coated LLM-105-based PBX could be improved by 27–52% and 47–100%, respectively. Furthermore, PBXs containing coated HMX and TATB also exhibited higher strength, stronger toughness, higher creep resistance, and higher microstructural thermal stability simultaneously, compared to PBXs containing pristine explosive crystals. This research group also adopted theoretical calculations to explain the accounts for mechanical enhancement, which combines the van

der Waals forces, π–π interaction, and hydrogen bonds, as well as surface roughness. Since the complex interfacial interaction cannot be experimentally observed, molecular dynamic simulation and quantitative analysis were widely used to demonstrate related mechanisms. Zeng and Lin [35,36] used different theoretical calculation methods to study the interfacial strength and contributing factors, and obtained the consistent conclusion that PDA can form a strong interfacial interaction with energetic crystals and crosslink with a polymeric matrix via abundant hydrogen bonds.

Another attempt to improve the mechanical properties is by grafting polymer binders onto explosive crystals directly through the abundant functional groups on PDA. Zeng et al. grafted three polymers including glycidyl azide polymer (GAP), polyethylene glycol (PEG), and polytetramethylene ether glycol (PTMEG) onto TATB via PDA, and PBXs prepared by these grafted explosive crystals showed remarkably increased mechanical properties; in particular, the PTMEG-grafted TATB exhibited excellent wettability of two phases [37]. Further, they grafted two hyperbranched polyesters (HBPs) onto TATB via hydroxyl groups on PDA [38]. PBXs using modified TATB showed improved storage modulus, creep resistance properties, and higher wettability, and tensile and compressive strength were increased significantly by 26.5% and 19.8%, respectively, due to the strong interfacial reinforcement of HBPs. This "grafting-from" route was popularized to tailor mechanical properties of other energetic polymeric composites by multilevel core–shell strategies [39], which resulted in a high-efficiency mechanical enhancement, including tensile and compressive strength and creep resistance.

### 3.3. Thermal, Sensitivity, and Safety Modification

Except for the mechanical enhancement of PBXs prepared from PDA-coated explosive crystals, the influences of PDA on the thermal, sensitivity, and safety properties of EMs also raised attention right from the start. Zhu et al. noticed that the friction and impact sensitivities of HMX@PDA particles were 30% and 50% lower, respectively, than those of HMX, along with displaying better wettability [40]. Another extension for reducing the sensitivities by PDA is constructing a "core–dual-shell (CDS)" structure with a completely inert material or an explosive exhibiting relatively lower sensitivity. Regarding HMX as a sensitive explosive, Lin et al. used TATB nanoparticles as the inner shell and PDA as the outer shell to fabricate an HMX@TATB@PDA CDS microstructure via a facile ultrasonic method and a simple immersion method, respectively [41]. Consistent with their previous works, PDA coating resulted in the increased β-δ phase transition temperature of HMX from 197.0 to 212.8 °C, and the impact sensitivity was 50% lower than that of the physical mixture without deteriorating its explosion performance; these properties were also combined with improved mechanical strength and roughness, storage modulus, and creep resistance. Instead of TATB, they also used high-melting-point paraffin wax (HPW) as the inner shell for CDS, which demonstrated a 117% increase in impact energy than that using conventional wax due to the high melting enthalpy of HPW and the associated optimization molding process, and a stronger interfacial interaction [42].

This CDS-sensitivity-reducing route also derives from those microstructures regarding PDA as an inner shell utilizing its adhesion effect. Amorphous $TiO_2$ was used to reduce the impact and electrostatic discharge sensitivities of RDX (hexahydro-1,3,5-trinitro-1,3,5-triazine); however, uniform deposition of $TiO_2$ as a shell to the RDX crystals surfaces was a great challenge. Wang et al. employed PDA as a bio-adhesive agent coated on the surface of RDX to enhance the interfacial adhesion between RDX and amorphous $TiO_2$, which improved the impact energy and electrostatic discharge energy by about 200.0% and 514.3% in the samples with optimal PDA and $TiO_2$ contents, respectively, compared with pristine RDX [43].

It was also found that this explosive@PDA@$TiO_2$ structure has the effect of enhancing the thermal decomposition of HMX. In Zhu's work, $TiO_2$ nanoparticles were anchored on the surface of HMX by PDA coating, which decreased the decomposition onset temperature and peak temperature by about 60 °C and 35 °C, respectively [44]. However, PDA itself

contributes to improving the thermal stability of EMs in many cases. Yu et al. coated LLM-105 with PDA and investigated its thermal decomposition behaviors in different heating conditions [45]. They found that PDA can hardly affect the thermal stability of LLM-105 under non-adiabatic conditions but remarkably intensified under adiabatic conditions. They proposed an inference that the amount of heat released by PDA decomposition and the heat-exchange rate between the sample and environment are the key to this effect, and it was confirmed by the Comsol numerical simulation in their work. Meanwhile, He's group constructed a multidimensional filler structure composed of 2D graphene nanoplatelets (GNPs), 0D AgNPs, and a bioinspired interfacial PDA layer on TATB, and PBXs were prepared from the referred-to TATB [46]. The thermal conductivity of PBXs was dramatically enhanced due to the stable 2D GNPs and 0D AgNPs anchored by PDA, which served as thermally conductive "bridges" to link the components and facilitate the heat transfer across the interfaces. Their previous work adopted carbon nanofillers, including multiwalled carbon nanotubes, graphene, and graphene nanoplates, into the surface of TATB via PDA anchoring, and the prepared PBXs exhibited improved thermal conductivity, as well as creep resistance and tensile and compression strength [47].

*3.4. Reactivity and Energetic Performances Tailoring*

Due to the impact of coating and its stable anchoring effect, a dramatic influence, reactivity and energetic performances tailoring of PDA was also observed during its applications in EMs. Yan's group focused on the performances tailoring of MICs by adopting PDA as a binding layer [10,48,49]. In the early stage of their research, PDA was used to coat aluminum nanoparticles (nAl) and poly(tetrafluoroethylene) for preparing MICs, which exhibiting increased energy release and reduced sensitivity, and more importantly tunable reactivity by controlling the thickness of PDA coating. After understanding the anchoring effect of PDA, they further constructed nAl@PDA@oxidizers MICs by the direction of PDA on the heterogeneous nucleation and growth of $Cu(OH)_2$ and CuO after nAl functionalization, as shown in Figure 3. The abovementioned MICs showed an improved initial reaction temperature, enhanced energy release, lower combustion temperature, and higher combustion efficiency compared with conventional MICs. The nAl functionalization provided more possibilities for constructing MICs containing different ingredients and tailoring their energy release processes. The results from similar works from Yan's group include introducing energetic metal–organic frameworks (MOF), which can decompose to metal oxide as oxidizers in conventional nanothermites and release extra energy during decomposition into MICs. These MICs undergo self-sustainable combustion with multilevel energy releases with lower ignition temperatures.

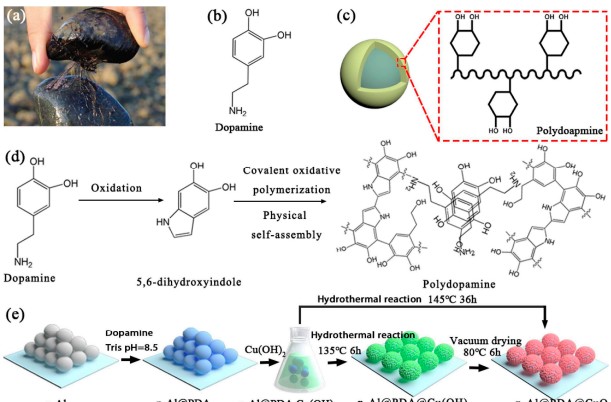

**Figure 3.** (**a**) A picture of a mussel, showing the strong adhesive of the byssus. (**b**) The molecular structure of dopamine. (**c**) The structure of polydopamine film binding on the particle surface. (**d**) The polymerization of dopamine. (**e**) Schematic description of the fabrication of n-Al@PDA@CuO MICs constructed by dopamine-nucleated crystal growth. Reprinted with permission from Ref. [48]. 2019, Elsevier.

Wang's group also attempted to improve the energetic performance through PDA when preparing CL-20-based energetic film by employing 3D micro-jet printing [50]. PDA was utilized as a linking bridge to induce the in situ self-assembly of CL-20-based polymeric composites. They suggested that PDA enhanced the physical entanglement between the binders and energetic crystal, and resulted in an improved detonation performance, elastic modulus, and deposition density.

## 4. Conclusions

To summarize, PDA plays an essential role in the crystal phase stability, thermal stability, and mechanical sensitivity of modified EMs, and its binding effect also provides more possibilities for mechanical enhancement, reactivity tailoring, and substances anchoring via secondary functionalization. This review firstly introduced possible self-polymerization mechanisms of dopamine and listed some representative methods to accelerate this process for inspiring researchers in the field of EMs, based on the fact that current accelerating methods in EMs still remain in adjusting concentrations of dopamine and oxygen, and reaction temperatures. Then we amply demonstrated the applications of PDA in EMs in terms of its effects, including surface modification of single energetic crystals; interfacial bonding and mechanical enhancement of energetic composites; thermal, sensitivity, and safety modification; as well as reactivity and energetic performances tailoring.

## 5. Future Directions

Although the applications of PDA in EMs are on the rise, some issues remain unaddressed in this field, which may be considered as future directions, or more appropriately, need to be solved. These are as follows:

1. The content of PDA, which is significant to the energetic performances of EMs, is very hard to measure due to its insolubility. Since the current method only can characterize the thickness of PDA film, converting thickness to content seems to be necessary.
2. Low polymerization and deposition rates of PDA on EMs limit its applications. PDA formation processes are affected by many factors including temperature, pH value, oxidizing catalysts, and so on. Optimal processing conditions are worth exploring, especially for EMs with high sensitivity, reactivity, or oxidability.
3. The binding effect or secondary functionalization of PDA could be further extended to combine substances with high surface tension to fabricate CDS microstructures which originally cannot bond stably.
4. With rapid development of EMs, the applications of PDA in the next-generation EMs such as $N_5$ salts, polyCO, and other high-nitrogen compounds should raise attention for stabilization and sensitivity reduction.

**Author Contributions:** Conceptualization, Q.J. and Y.O.; validation, S.D.; writing—original draft preparation, Z.Q.; writing—review and editing, D.L.; supervision, Y.O.; funding acquisition, Y.O.; resources, J.P. and P.L. All authors have read and agreed to the published version of the manuscript.

**Funding:** This work was funded by the National Natural Science Foundation of China (Grant No. 22005031) and the Autonomous Research Program of SKLEST (Grant No. QNKT22-13).

**Data Availability Statement:** The data presented in this study are available on request from the corresponding author.

**Conflicts of Interest:** The authors declare no conflict of interest.

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
