# Peer review of "Recent Advances in Polydopamine for Surface Modification and Enhancement of Energetic Materials: A Mini-Review"

_crystals, doi:10.3390/cryst13060976_

Round 1

Reviewer 1 Report

The review of recent advances in polydopamine for surface modification and enhancement of energetic crystal, from my point of view, is well organized and written. It is an interesting topic. 

The authors have to review the references 11 and 12, do not correspond to authors mentioned in the text.

Please, mention the contribution of each author in the preparation of this manuscript. 

I found the manuscript interesting and well done, it can be published after making the little suggested corrections.

Author Response

Dear Reviewer,

We’ like to express our gratitude firstly for your careful review and valuable comments, and we have revised our manuscript strictly according to your comments. Also, I, myself must apologize for the inadvertent mistake in the wrong order of references, it is corrected now.

  1. The references are all checked, and they’re all in the right sites.
  2. We have added the Contributions of Authors, Data Availability Statement and Conflicts of Interest in our revised manuscript.

Kind regards

Dr. Yapeng Ou

SKLEST, BIT

Reviewer 2 Report

The manuscript entitled “Recent Advances in Polydopamine for Surface Modification and Enhancement of Energetic Crystals” reported a concise and comprehensive review of recent progresses in researches and studies of polydopamine (PDA) for modifications of energetic materials (EM), as well as a brief perspective on future directions of PDA applications in EM. The authors have presented an ordered and integrated mini review to the audience how PDA can be potentially applied to tune mechanical, thermal, sensitivity and safety properties of traditional EM. The limitations have also been discussed. The reviewer recommends this work can be considered for publication for Crystals with the following issues to be stressed.

1. The effects of modelling and simulations in the development of PDA surface modification of energetic crystals and other related property modifications have not been discussed. It is suggested that the authors should add more paragraphs to present if technology of modelling and simulations could play roles in this field.

2. Outlooks of surface modification of nest generation EM by PDA should also be discussed, such as for N5+ and N5- salts.

3. The sections of Author Contributions, Data Availability Statement, Acknowledgments and Conflicts of Interest were not well written.

 Minor editing of English language required

Author Response

Dear Reviewer,

We’ like to express our gratitude for your careful review and valuable suggestions. We have revised our manuscript strictly according to your comments, and your comments were addressed as below.

  1. The effects of modelling and simulations in the development of PDA surface modification of energetic crystals and other related property modifications have not been discussed. It is suggested that the authors should add more paragraphs to present if technology of modelling and simulations could play roles in this field.

Respond: It’s a very helpful advice to improve the quality of our work. We ignored this part when preparing our review, and in fact, many references we cited do use modelling and simulation to explain the mechanisms of experimental results.

Our manuscript was prepared according to structure that the effects of PDA on EMs, so we have added the contents of simulation and also some extra references in each section such as the mechanical enhancement and thermal modification

(eg. In Section 3.2, “This research group also adopted theoretical calculations to explain the accounts for mechanical enhancement, which combines the van der Waals forces, π–π interaction, and hydrogen bonds, as well as surface roughness. Since the complex interfacial in-teraction cannot be experimentally observed, molecular dynamic simulation and quantitative analysis were widely used to demonstrate related mechanisms. Zeng and Lin [33,34] used different theoretical calculation methods to study the interfacial strength and contributing factors, and obtained consistent conclusion that PDA can form strong interfacial interaction with energetic crystals and crosslink with poly-meric matrix via abundant hydrogen bonds.”

In Section 3.3, “Yu et al. coated LLM-105 with PDA, and investigated its thermal decomposition be-haviors in different heating conditions [41]. They found that PDA can hardly affect the thermal stability of LLM-105 under non-adiabatic conditions but remarkably in-tensified under adiabatic conditions. They proposed an inference that the amount of heat released by PDA decomposition and the heat-exchange rate between sample and environment are the key to this effect, and it was confirmed by the Comsol numerical simulation in their work”.

  1. Outlooks of surface modification of nest generation EM by PDA should also be discussed, such as for N5+ and N5- salts.

Respond: We have to admit that we cannot find related works about PDA modified next generation EMs when we collected references. But your suggestion is very meaningful, and we have added some discussions in Future Directions as “With rapid development of EMs, the applications of PDA in the next generation EMs such as N5 salts, polyCO and other high nitrogen compounds should raise attention for stabilization and sensitivity reduction”.

  1. The sections of Author Contributions, Data Availability Statement, Acknowledgments and Conflicts of Interest were not well written.

Respond: We must apologize for this mistake in preparation of our manuscript. We have added the Contributions of Authors, Data Availability Statement and Conflicts of Interest in our revised manuscript.

Kind regards

Dr. Yapeng Ou

SKLEST, BIT

Reviewer 3 Report

This review is devoted to recent developments in application of dopamine energetic materials. Hovewer, it's not clear what author exactly mean with "energetic materials"  or "energetic composites" and what applications are related with this area. Thus, a subsection about definition and features of energetic materials is required and clarification in abstract. Classification of the main part is required reassembly (for example, based on exact application or based on dopamine role (coating, inclusion or other). Alternatively, to clarify suggested by authors classification additional figure or scheme could be introduced. Also some important works in the field of dopamine coatings are missing 10.1016/j.porgcoat.2022.107359, 10.1002/pi.6358. Recommnedation is reconsider after major revision.

The quality of English is acceptable

Author Response

Dear Reviewer,

We’ like to express our gratitude firstly for your careful review and valuable comments, and we have revised our manuscript strictly according to your comments. We have addressed your concerns in our manuscript, and the point-by-point answers are listed below.

  1. it's not clear what author exactly mean with "energetic materials"  or "energetic composites" and what applications are related with this area. Thus, a subsection about definition and features of energetic materials is required and clarification in abstract.

Respond: This suggestion is very helpful for improving the readability of our work. We have added a definition in Introduction as Energetic materials (EMs) are very important power resources for civilian and military applications covering explosives, propellants and pyrotechnics, which release energy in the forms of combustion and explosion”. Energetic Materials (EMs) is a broad concept including energetic crystals and energetic composites, which covers simple substances (eg. compound explosives, metallic fuels and oxidizer) and composites (eg. mixed civilian and military explosives, propellants and pyrotechnics).

  1. Classification of the main part is required reassembly (for example, based on exact application or based on dopamine role (coating, inclusion or other). Alternatively, to clarify suggested by authors classification additional figure or scheme could be introduced.

Respond: Thank you for your suggestion. We have added a Graphical Abstract to clearly demonstrate the structure of our manuscript. The main aim of this work, is to summarize the effects and applications of PDA in energetic materials, including the compact coating on the surface of energetic crystals (simple substances) for crystal phase, thermal stability, and interfacial interaction enhancement in energetic composites (especially for composites containing both polymeric matrix and energetic crystals), and some other effects like reaction rate tailoring. Also, before that, the possible polymerization mechanism of PDA was also briefly introduced.

  1. Also some important works in the field of dopamine coatings are missing 10.1016/j.porgcoat.2022.107359, 10.1002/pi.6358.

Respond: We must apologize for our negligence in collecting related works and references, your suggested works were cited in our work as Reference [3] and [6]. These two works are newly reported after we finishing works acquisition.

Kind regards

Dr. Yapeng Ou

SKLEST, BIT

Reviewer 4 Report

This mini review is publishable, but some sentences are confusing. I accept its publication after minor English correction. 

Dear Editor, 
I have reviewed the MS and I think its publishable after some minor English correction. 

Anshuman 

Author Response

Dear Reviewer,

We’ like to express our gratitude firstly for your careful review and valuable comments.

We have throughly checked and modified our English expressions according to the suggestions from native English speakers. I believe our manuscript is now clear and friendly to readers.

Kind regards

Dr. Yapeng Ou

SKLEST, BIT